# Water-mediated recognition of t1-adenosine anchors Argonaute2 to microRNA targets

Nicole T Schirle[1], Jessica Sheu-Gruttadauria[1], Stanley D Chandradoss[2], Chirlmin Joo[2]*, Ian J MacRae[1]*

[1]Department of Integrative Structural and Computational Biology, The Scripps Research Institute, La Jolla, United States; [2]Kavli Institute of NanoScience, Department of BioNanoScience, Delft University of Technology, Delft, Netherlands

**Abstract** MicroRNAs (miRNAs) direct post-transcriptional regulation of human genes by guiding Argonaute proteins to complementary sites in messenger RNAs (mRNAs) targeted for repression. An enigmatic feature of many conserved mammalian miRNA target sites is that an adenosine (A) nucleotide opposite miRNA nucleotide-1 confers enhanced target repression independently of base pairing potential to the miRNA. In this study, we show that human Argonaute2 (Ago2) possesses a solvated surface pocket that specifically binds adenine nucleobases in the 1 position (t1) of target RNAs. t1A nucleotides are recognized indirectly through a hydrogen-bonding network of water molecules that preferentially interacts with the N6 amine on adenine. t1A nucleotides are not utilized during the initial binding of Ago2 to its target, but instead function by increasing the dwell time on target RNA. We also show that N6 adenosine methylation blocks t1A recognition, revealing a possible mechanism for modulation of miRNA target site potency.

*For correspondence:
C.Joo@tudelft.nl (CJ);
macrae@scripps.edu (IJM)

**Competing interests:** The
authors declare that no
competing interests exist.

**Reviewing editor:** Phillip D
Zamore, Howard Hughes Medical
Institute, University of
Massachusetts Medical School,
United States

## Introduction

MicroRNAs (miRNAs) are an abundant class of regulatory molecules with diverse biological functions in plants and animals (*Lagos-Quintana et al., 2001*; *Lau et al., 2001*; *Lee and Ambros, 2001*; *Reinhart et al., 2002*). Over two thousand unique human miRNA sequences have been reported and more than half of all protein-coding human genes are predicted to contain a conserved miRNA recognition site (*Friedman et al., 2009*; *Kozomara and Griffiths-Jones, 2011*). miRNAs function as guides for Argonaute proteins, which use the sequence information encoded in each miRNA to identify complementary sties in mRNAs targeted for repression (*Liu et al., 2004*; *Meister et al., 2004*). Argonautes then recruit additional silencing factors that mediate translational repression and degradation of the targeted mRNAs (*Huntzinger and Izaurralde, 2011*).

Pioneering studies in the nematode *Caenorhabditis elegans* showed that perfect complementarity between miRNAs and their targets is not necessary for silencing (*Lee et al., 1993*; *Wightman et al., 1993*). Examination of regulatory elements in fly mRNAs then revealed a striking degree of complementarity to the 5′ ends of subset of conserved miRNAs, suggesting that base paring interactions with nucleotides towards the 5′ end of the miRNA are particularly important for target recognition (*Lai, 2002*). Indeed, phylogenetic analysis showed that pairing to the miRNA 'seed region' (nt 2–7 or 2–8, from the miRNA 5′ end) is the most evolutionarily conserved feature of miRNA targets in animals (*Lewis et al., 2003*; *Brennecke et al., 2005*; *Krek et al., 2005*; *Lewis et al., 2005*), and complementarity to the miRNA seed is generally sufficient to elicit significant levels of target recognition and repression (*Doench and Sharp, 2004*; *Brennecke et al., 2005*; *Lim et al., 2005*). Careful comparison of miRNA seed-matched sites in the 3′ untranslated regions (UTRs) of human, mouse, rat, dog, and chicken genomes also revealed that an adenosine (A) nucleotide opposite miRNA nucleotide-1 is a conserved feature of many vertebrate

**eLife digest** Stretches of DNA known as genes provide the instructions to make the proteins and RNA molecules a cell needs to work. To make a protein, the gene is used as a template to make a type of RNA molecule called messenger RNA (mRNA), which is subsequently 'translated' into a protein. Most genes do not need to produce proteins all of the time, and so cells have several ways of stopping proteins from being made. For example, the Argonaute family of proteins prevents mRNA molecules from being translated into proteins.

Argonautes are guided to their targets by short RNA molecules called microRNAs. RNA molecules are made up of a sequence of building blocks known as nucleotides, each of which can only bind to one other type of nucleotide. If part of the nucleotide sequence of a microRNA molecule corresponds with part of the nucleotide sequence of the mRNA, the two RNA molecules will bind to each other. This enables the microRNA and the Argonaute protein to prevent the mRNA being translated. If the mRNA has an adenine nucleotide in a particular position (called 't1') near the binding region in the mRNA sequence, Argonaute proteins will prevent translation more effectively. An adenine nucleotide in the t1 position is also known as a t1A nucleotide.

In 2014, researchers revealed the structure of a human Argonaute protein called Argonaute2 when it is bound to a microRNA-mRNA pair. This revealed that t1A nucleotides—but not other nucleotide types in the t1 position—interact with a 'pocket' in the Argonaute protein. However, it was not clear how the adenine nucleotide is recognized.

Now, Schirle et al.—including several of the researchers involved in the 2014 work—use a technique called X-ray crystallography to examine how the t1A nucleotide interacts with Argonaute2 in more detail. This revealed that the Argonaute2 pocket contains many water molecules that form an organized network. This network interacts with part of the t1A nucleotide and helps to lock Argonaute2 onto its microRNA target sites. The discovery of the pocket and how t1A is recognized may now be used to design more effective 'anti-miRs'—synthetic microRNA inhibitors that can treat diseases in which microRNAs work incorrectly, a feature common to many forms of cancer.

miRNA target sites (*Lewis et al., 2005*). These t1A nucleotides confer enhanced repression of miRNA targets beyond pairing to the seed region alone (*Grimson et al., 2007*; *Nielsen et al., 2007*; *Baek et al., 2008*; *Selbach et al., 2008*). Curiously, unlike the seed-matched region, t1A nucleotides function independently of the identity of miRNA nucleotide-1, indicating that t1A recognition occurs through a mechanism that is distinct from base pairing with the miRNA guide (*Grimson et al., 2007*; *Nielsen et al., 2007*; *Baek et al., 2008*).

We recently reported crystal structures of human Argonaute2 (Ago2) bound to a guide RNA and short, seed-paired target RNAs (*Schirle et al., 2014*). These structures revealed that Ago2 cradles the duplex formed by miRNA nucleotides 2–7 and the complementary target RNA, explaining why seed pairing is critical for recognition of miRNA target sites (*Figure 1A*). We also noted that t1A nucleotides interact with a surface pocket formed at the interface of the L2 and MID domains of Ago2, and proposed that this additional interaction may contribute to the affinity of Ago2 for miRNA target sites. However, the t1-nucleotide binding pocket is large enough to accommodate any of the four natural RNA bases, and the contacts between Ago2 and t1A appear to be almost entirely non-specific (*Figure 1B*). Therefore, it is not clear how A nucleotides in the t1 position of miRNA targets are recognized and confer enhanced repression. In this study, we show that the t1-nucleotide binding pocket in Ago2 preferentially interacts with A nucleotides through water-mediated contacts to adenosine N6 amine. Adding a methyl group to the t1A N6 amine reduces target affinity, raising the possibility that adenosine methylation could, in principle, lead to partial derepression of miRNA targets containing 7mer-A1 or 8mer sites. We also present data indicating that t1A is not used in the initial search for target sites, but instead provides an anchor that helps retain Ago2 on seed-matched sites on target RNAs.

## Results

### An adenosine-specific t1-binding pocket at the MID-L2 interface

We first visualized how Ago2 engages non-A t1 nucleotides by determining crystal structures of Ago2 bound to short target RNAs with cytosine (C), uracil (U), or guanine (G) in the t1 position (*Table 1*). The

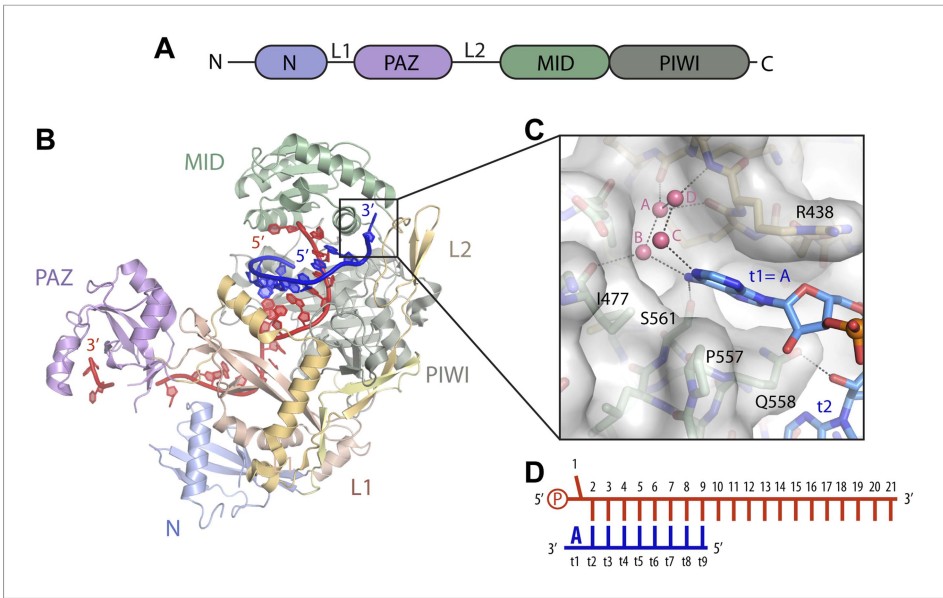

**Figure 1**. Structure of the t1 nucleotide binding pocket. (**A**) Linear schematic of the Argonaute2 (Ago2) primary structure. (**B**) Crystal structure of Ago2 bound to a guide RNA (red) and target RNA bearing t1A nucleotide (blue; PDB ID: 4W5O). (**C**) Close-up view of the t1-binding pocket. Ordered water molecules shown as pink spheres. Protein shown in stick and surface representations. Target RNA shown as sticks. (**D**) Linear schematic of crystallized guide and target RNAs.

overall conformation of Ago2 is nearly identical in the four different t1 structures, indicating that identity of the t1 nucleotide does not affect the structure of the protein. Inspection of target RNA omit maps, in which the target RNAs were excluded from refinement, revealed little or no well-defined electron density for all of the non-A nucleotides, indicating that the non-A t1 nucleotides were mostly disordered (*Figure 2*). In contrast, the target omit map from the t1A data set had clearly defined density for the t1A inside the t1-nucleotide binding pocket. Moreover, although residual electron density was observed for t1G and t1C nucleotides, binding experiments using the same target RNAs show that these interactions do not contribute substantially to target affinity (*Figure 2*). We conclude that, although the t1-binding site is large enough to accommodate any of the four natural RNA bases, adenine is the only nucleobase that associates stably enough to contribute to overall target affinity.

## An unmodified purine N6 amine is required for t1 recognition

The contacts between Ago2 and t1A appear to be largely non-specific (*Figure 1*). Therefore, the mechanism by which Ago2 distinguishes the t1A purine ring from t1G is unclear. Two major differences between adenine and guanine are: position-6, where adenine has an exocyclic amine, and guanine has a carbonyl; and position-2, where guanine has an exocyclic amine, and adenine has no substituent group (*Figure 3A*). We dissected the influence of these two elements on t1-binding by examining interactions with target RNAs with either inosine (I) or 2,6-diaminopurine (DAP) in the t1 position. DAP has an N6 amine (like adenine), and an N2 amine (like guanine). Conversely, inosine has an O6 carbonyl (like guanine), but lacks an N2 amine (like adenine). Ago2 bound the t1I target with an affinity similar to t1G (*Table 2*) and no electron density was observed for the t1I nucleotide target omit maps, indicating that removal of the guanine N2 amine is insufficient to promote interactions with the t1-binding site (*Figure 3B*). In contrast, we observed unambiguous electron density for the t1DAP nucleotide, which bound Ago2 in the same position and *anti* conformation as t1A (*Figure 3C*). Moreover, the affinity of the t1DAP target is ~1.7-fold greater than that of the equivalent t1A target, indicating that Ago2 has a modest, though significant, preference for t1DAP over t1A (p-value = 0.0025; two-tailed, unpaired Student's *t*-test). We conclude that the purine N6 and N2 amines both have positive effects on t1-binding, and that the N6 amine is the major determinant for distinguishing the t1A purine ring from tG. Consistent

**Table 1**. Crystallographic and refinement statistics for wild-type Ago2-guide-target complexes

| Target RNA | t1-C | t1-G | t1-U | t1-DAP | t1-Inosine |
|---|---|---|---|---|---|
| PDB code | 4Z4C | 4Z4D | 4Z4E | 4Z4F | 4Z4G |
| Space group | $P12_11$ | $P12_11$ | $P12_11$ | $P12_11$ | $P12_11$ |
| Unit cell dimensions | | | | | |
| $a$ (Å) | 55.69 | 55.74 | 55.64 | 55.86 | 55.66 |
| $b$ (Å) | 116.56 | 117.02 | 116.84 | 116.60 | 117.0 |
| $c$ (Å) | 69.61 | 69.87 | 69.74 | 70.38 | 70.1 |
| $\beta$ (°) | 92.43 | 92.43 | 92.43 | 92.52 | 92.40 |
| Ago2 per ASU | 1 | 1 | 1 | 1 | 1 |
| Data collection | | | | | |
| Wavelength (Å) | 0.97945 | 0.97950 | 0.97918 | 0.97950 | 0.97950 |
| Resolution (Å) | 38.85–2.30 (2.38–2.30) | 39.01–1.60 (1.63–1.60) | 55.60–1.80 (1.90–1.80) | 38.86–2.80 (2.95–2.80) | 39.01–2.70 (2.83–2.70) |
| Total reflections | 133,678 | 532,622 | 351,634 | 65,587 | 75,249 |
| Unique reflections | 38,614 | 113775 | 82,071 | 21,078 | 24,061 |
| Completeness (%) | 98.4 (96.3) | 96.8 (93.5) | 99.7 (99.7) | 95.0 (92.6) | 97.2 (91.9) |
| Redundancy | 3.5 (3.4) | 4.7 (4.6) | 4.3 (3.7) | 3.1 (3.0) | 3.1 (2.9) |
| I/σI | 13.1 (2.2) | 13.7 (2.0) | 10.3 (1.9) | 9.9 (2.4) | 9.6 (2.2) |
| $R_{merge}$ | 7.7 (53.0) | 5.5 (74.9) | 9.8 (81.9) | 9.8 (57.7) | 8.4 (48.2) |
| $R_{pim}$ | 7.3 (49.2) | 3.1 (59.0) | 5.3 (47.4) | 9.7 (57.1) | 6.4 (36.3) |
| Refinement | | | | | |
| Resolution (Å) | 35.30–2.30 | 39.01–1.60 | 40.27–1.80 | 35.47–2.80 | 35.41–2.70 |
| R-free/R-factor | 21.86/16.93 | 18.90/16.31 | 18.54/15.75 | 23.30/18.39 | 22.44/17.86 |
| R.M.S. deviation | | | | | |
| Bond distances (Å) | 0.014 | 0.014 | 0.007 | 0.006 | 0.008 |
| Bond angles (°) | 1.454 | 1.460 | 1.131 | 0.903 | 0.989 |
| Number of atoms | | | | | |
| Non-hydrogen, protein | 6429 | 6469 | 6429 | 6421 | 6404 |
| Non-hydrogen, RNA | 568 | 580 | 552 | 571 | 572 |
| Phenol | 28 | 28 | 28 | 21 | 28 |
| Isopropanol | 4 | 20 | 8 | 0 | 0 |
| Phosphate | 10 | 0 | 5 | 0 | 0 |
| Mg | 3 | 3 | 3 | 3 | 3 |
| Water | 279 | 554 | 545 | 16 | 105 |
| Ramachandran Plot | | | | | |
| Most favored regions | 95.33% | 96.79% | 96.97% | 94.93% | 95.17% |
| Additionally allowed | 4.54% | 3.21% | 3.03% | 4.94% | 4.83% |
| Generously llowed | 0.13% | 0.00% | 0.00% | 0.13% | 0.00% |

with this idea, adding a methyl group to the t1A N6 amine reduced target affinity to that of a non-t1A target (*Figure 3D*).

## An organized water network inside the t1-binding pocket

The only direct contact between the t1A N6 amine and Ago2 is a hydrogen bond to the side chain of Ser-561 (*Figure 1*). Because the serine alcohol can function as both a hydrogen bond acceptor and donor, it is unlikely that this interaction alone can discriminate a purine N6 amine from an O6 carbonyl.

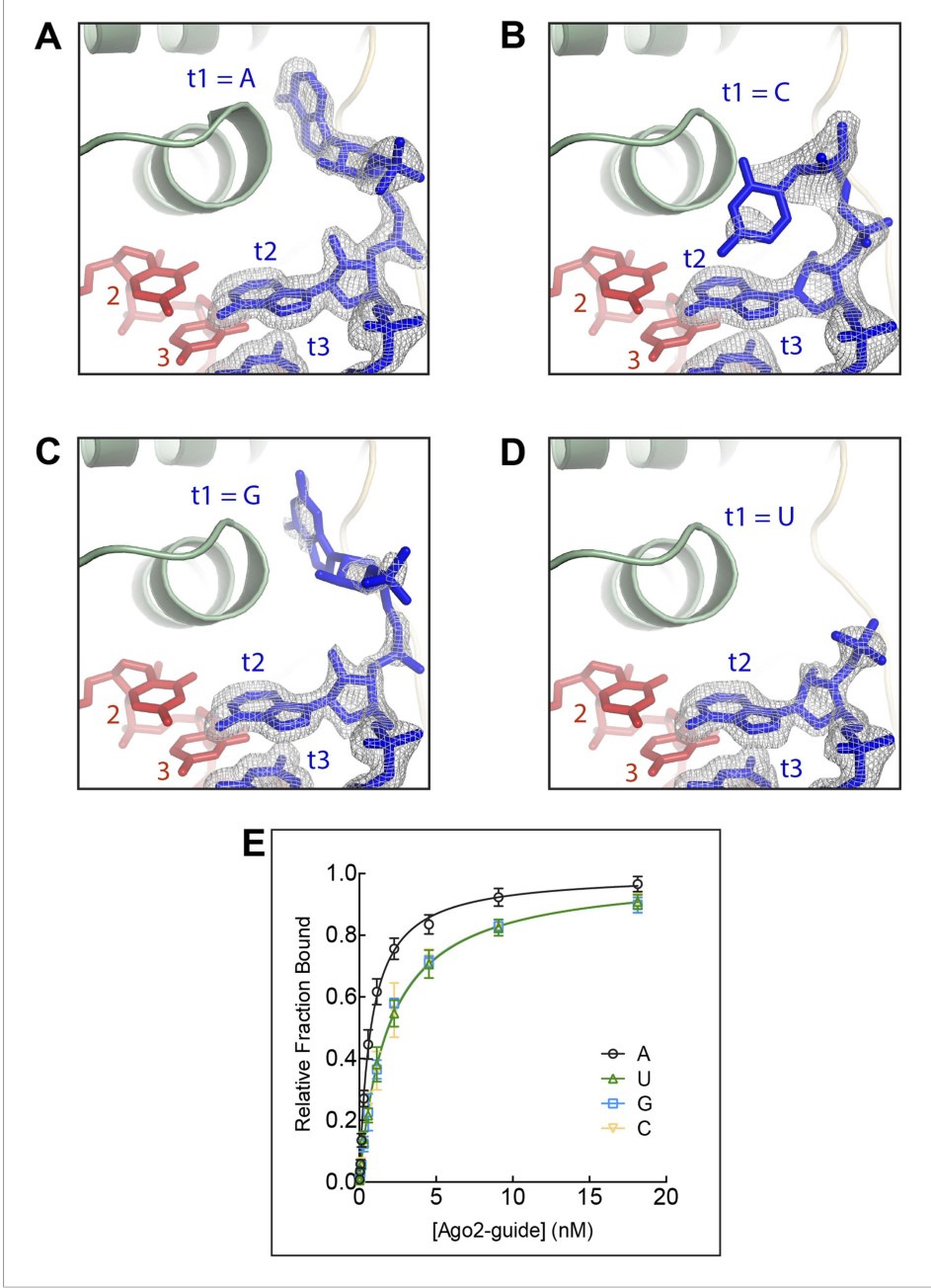

**Figure 2**. The t1 nucleotide binding pocket specifically recognizes adenosine Strong electron density was observed for adenosine in the t1-binding pocket, but not t1C, G, or U. (**A**) T1A target RNA omit map contoured at 3σ (grey mesh). (**B**) T1C target RNA omit map contoured at 3σ (grey mesh). (**C**) T1G target RNA omit map contoured at 3σ (grey mesh). (**D**) T1U target RNA omit map contoured at 3σ (grey mesh). (**E**) Plot of target bound vs Ago2-guide concentration for target RNAs with different t1-nucleotides.

Inspection of the atoms surrounding Ser-561 revealed no set of interactions that could conceivably steer the hydrogen-bonding capacity of the serine alcohol. However, examination of the solvent surrounding the t1A nucleobase identified four ordered water molecules in a hydrogen-bonding network that could selectively interact with purine N6 amines. In this model, both hydrogen atoms in water 'A' are engaged in hydrogen bonds with the main chain carbonyls of Lys-440 and Met-437 (and possibly the main chain carbonyl of Asp-480), leaving only lone pair electrons available for interactions with water 'B' (*Figure 4B*). Water B must therefore donate one hydrogen atom to establish a hydrogen

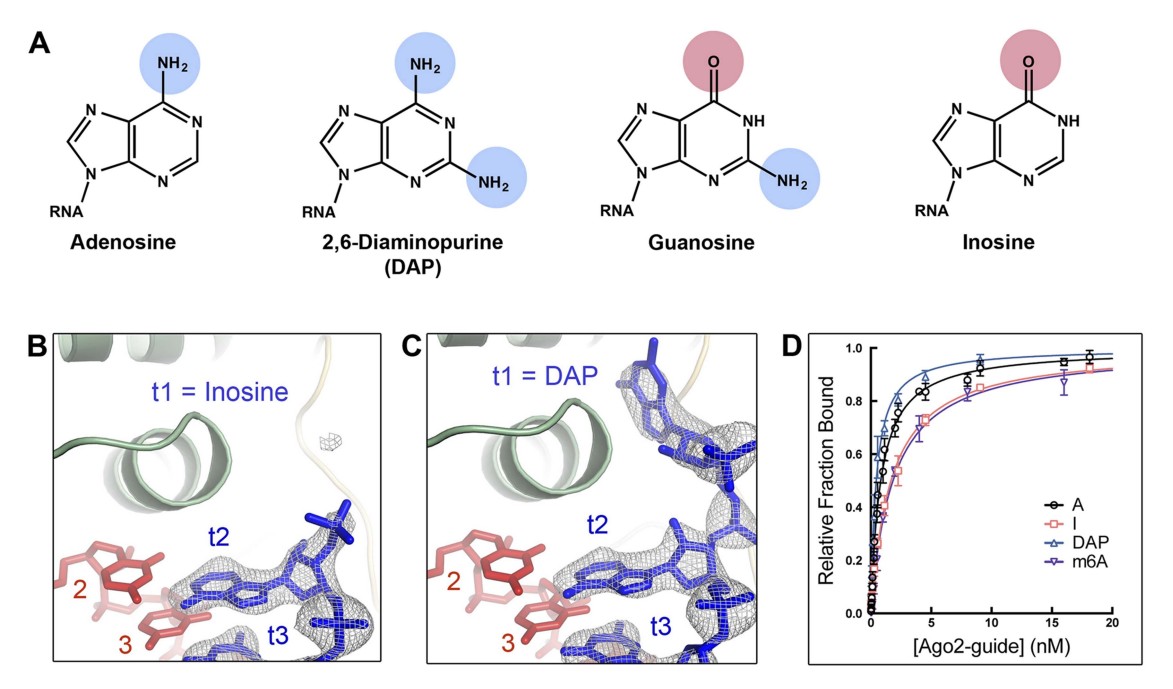

**Figure 3**. A purine N6 amine is required for t1 nucleotide recognition. (**A**) Chemical structures of adenosine, 2,6-diaminopurine (DAP), guanosine, and inosine (I) nucelobases. Hydrogen bond donors highlighted in blue; hydrogen bond acceptors highlighted in pink. (**B**) t1I target RNA omit map contoured at 3σ (grey mesh). No electron density was observed for t1I. (**C**) t1DAP target RNA omit map contoured at 3σ (grey mesh). Clear electron density was observed for t1DAP. (**D**) Plot of target bound vs Ago2-guide concentration for t1A, t1I, t1DAP, and t1m6A target RNAs.

bond with water A. The second hydrogen atom in water B is likely involved in a hydrogen bond with the main chain carbonyl of Ile-477. Thus, our model suggests that Ago2 uses main chain carbonyls inside the t1-binding pocket to direct the hydrogen atoms on water B away from the purine 6-position, leaving only lone pair electrons available for establishing interactions with t1 nucleotides. In the proposed orientation, the lone pair electrons on water B are positioned to accept a hydrogen bond from purine N6 amines and to repel O6 carbonyls. Waters 'C' and 'D' likely provide an additional layer of selectivity through interactions with the unprotonated N1 amine on t1A, and may interact with the purine N2 amine, which does not directly contact Ago2.

**Table 2**. Ago2-target affinities

| t1 nucleotide | $K_D$ (nM) | |
| --- | --- | --- |
| | WT Ago2 | A481T-Ago2 |
| A | 0.75 ± 0.04 | 1.5 ± 0.13 |
| G | 1.9 ± 0.09 | 1.8 ± 0.11 |
| U | 1.9 ± 0.10 | – |
| C | 1.8 ± 0.12 | – |
| DAP | 0.45 ± 0.03 | – |
| I | 1.7 ± 0.09 | – |
| m6A | 1.8 ± 0.12 | – |

Dissociation constants for wild-type (WT) and mutant Ago2 binding short target RNAs with different t1 nucleotides.

## Disruption of the water network extinguishes t1A recognition

We tested the role of water molecules in t1A recognition by designing a mutation in Ago2 that disrupts the water network inside the t1-binding site. Ala-481, which resides in the back of the t1-binding pocket and is within van der Waals contact distance of water B, was mutated to threonine. Our rational was that the A481T mutation would alter the hydrogen-bonding landscape of the t1-binding pocket and thereby perturb placement of the associated water molecules.

We first assessed the effects of the A481T mutation on Ago2 by determining the crystal structure of the mutant protein bound to a short target RNA bearing a t1A nucleotide (**Table 3**).

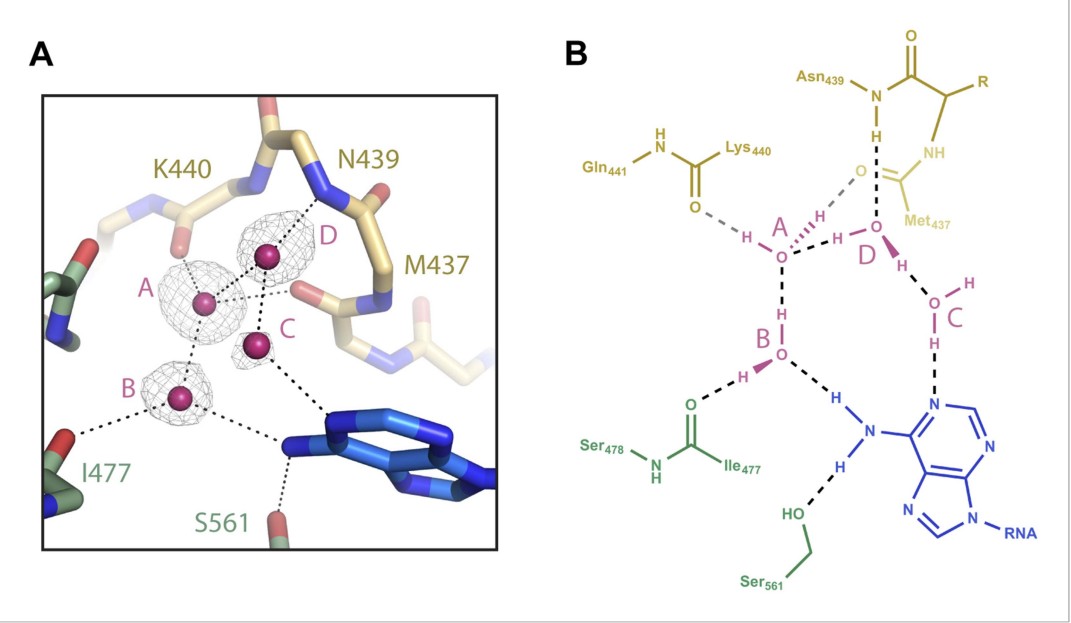

**Figure 4**. Water-mediated recognition of t1A. (**A**) Water network within the t1-binding pocket. Protein main chain shown as sticks, with side-chains (except S561) hidden for clarity. Water molecules shown as pink spheres. Water omit map shown contoured at 2.5σ (grey mesh). Potential hydrogen bonds shown as black dashed lines; t1A shown as blue sticks. (**B**) Flattened cartoon schematic of image shown in (**A**). Drawing illustrates proposed positions of hydrogen atoms within the t1A recognition network.

The overall structure of A481T Ago2 is nearly identical to the wild type protein, and the structure of the t1-binding pocket is almost completely unaffected by the A481T mutation. The Thr-481 side chain is in the same orientation as that of Ala-481, with the Cγ methyl pointed towards the protein interior and the Oγ alcohol extending into the t1-binding site (*Figure 5A*). Thus, the A481T mutation adds a hydrogen-bond acceptor/donor to the back of the t1-binding pocket without otherwise perturbing the structure of Ago2.

Although the structure of the A481T Ago2 is nearly identical to wild type, t1A recognition is severely impaired. Only residual electron density was observed for the t1A nucleotide in the target RNA omit map (*Figure 5B*). For comparison, we determined the structure of the A481T mutant bound to a target bearing a t1G nucleotide. Again, the structure of the mutant protein was essentially indistinguishable from the wild type and only residual electron density was observed for the t1G (*Figure 5C*). Moreover, the A481T Ago2 bound t1A and t1G target RNAs with nearly identical affinities (*Figure 5D*). Importantly, the Oγ hydroxyl of Thr-481 is 4.2 Å from the position of the t1A N6 amine in the wild-type structure, strongly indicating that the mutation affects t1A recognition indirectly (*Figure 5E*). We conclude that binding specificity in the t1-pocket is achieved by Ago2-directed solvent interactions with the t1 adenosine.

## t1A interactions increase the dwell time of Ago2 on miRNA target sites

Binding t1A nucleotides may increase the affinity of Ago2 for target RNAs by increasing the association rate, decreasing the rate of dissociation, or both. To distinguish between these possibilities, we used a single-molecule technique for directly observing Ago2-target binding events that we recently developed (*Chandradoss et al., 2015*). Briefly, Ago2 is loaded with a Cy3-labeled miRNA and introduced into a microfluidic chamber containing immobilized, Cy5-labled target RNAs. Pairing between the miRNA and the complementary site on the target RNA brings the Cy3 donor and Cy5 acceptor fluorophores into close proximity, leading to high Forster resonance energy transfer (FRET), which is observed at single-molecule resolution by total-internal-reflection microscopy (*Figure 6A–C*).

**Table 3**. Crystallographic and refinement statistics for mutant (A481T) Ago2-guide-target complexes

| Target RNA | t1-A | t1-G |
|---|---|---|
| PDB code | 4Z4H | 4Z4I |
| Space group | P12$_1$1 | P12$_1$1 |
| Unit cell dimensions | | |
| a (Å) | 55.69 | 55.60 |
| b (Å) | 116.60 | 116.60 |
| c (Å) | 70.10 | 69.62 |
| β (°) | 92.29 | 92.42 |
| Ago2 per ASU | 1 | 1 |
| Data collection | | |
| Wavelength (Å) | 0.99999 | 0.99999 |
| Resolution (Å) | 44.81–2.50 (2.61–2.50) | 44.69–2.80 (2.95–2.80) |
| Total reflections | 113332 | 85,923 |
| Unique reflections | 30,411 | 21,701 |
| Completeness (%) | 98.6 (98.3) | 99.2 (97.6) |
| Redundancy | 3.7 (3.8) | 4.0 (4.0) |
| I/σI | 9.2 (2.3) | 8.7 (2.4) |
| R$_{merge}$ | 12.7 (69.7) | 13.1 (60.0) |
| R$_{pim}$ | 12.0 (65.0) | 12.0 (55.0) |
| Refinement | | |
| Resolution (Å) | 44.81–2.50 | 44.69–2.80 |
| R$_{free}$/R$_{factor}$ | 21.13/17.21 | 23.32/19.18 |
| R.M.S. deviation | | |
| Bond distances (Å) | 0.008 | 0.005 |
| Bond angles (°) | 0.940 | 0.865 |
| Number of atoms | | |
| Non-hydrogen, protein | 6432 | 6412 |
| Non-hydrogen, RNA | 568 | 568 |
| Phenol | 28 | 21 |
| Isopropanol | 4 | 20 |
| Phosphate | 10 | 0 |
| Mg | 3 | 3 |
| Water | 135 | 23 |
| Ramachandran plot | | |
| Most favored regions | 95.19% | 94.67% |
| Additionally allowed | 4.68% | 5.33% |
| Generously allowed | 0.13% | 0.00% |

We first determined the rates at which Ago2 associates with target RNAs containing six nucleotides of complementarity to the guide seed region (nt 2–7) and either a t1U, t1A, or t1DAP nucleotide. Measuring the time of the first arrival revealed that Ago2 recognizes the three targets at similar rates ($k_{on(obs)}$ = 0.040 ± 0.009, 0.033 ± 0.004, and 0.041 ± 0.008 s$^{-1}$ nM$^{-1}$ for t1U, t1A, and t1DAP targets, respectively), demonstrating that t1-nucleotide identity has little, if any, effect on the association of Ago2 with target RNAs (*Figure 6D*). In contrast, the average dwell time (Δτ) of Ago2 on the t1A and t1DAP targets was more than threefold longer than on the t1U target (*Figure 6E*). The combined

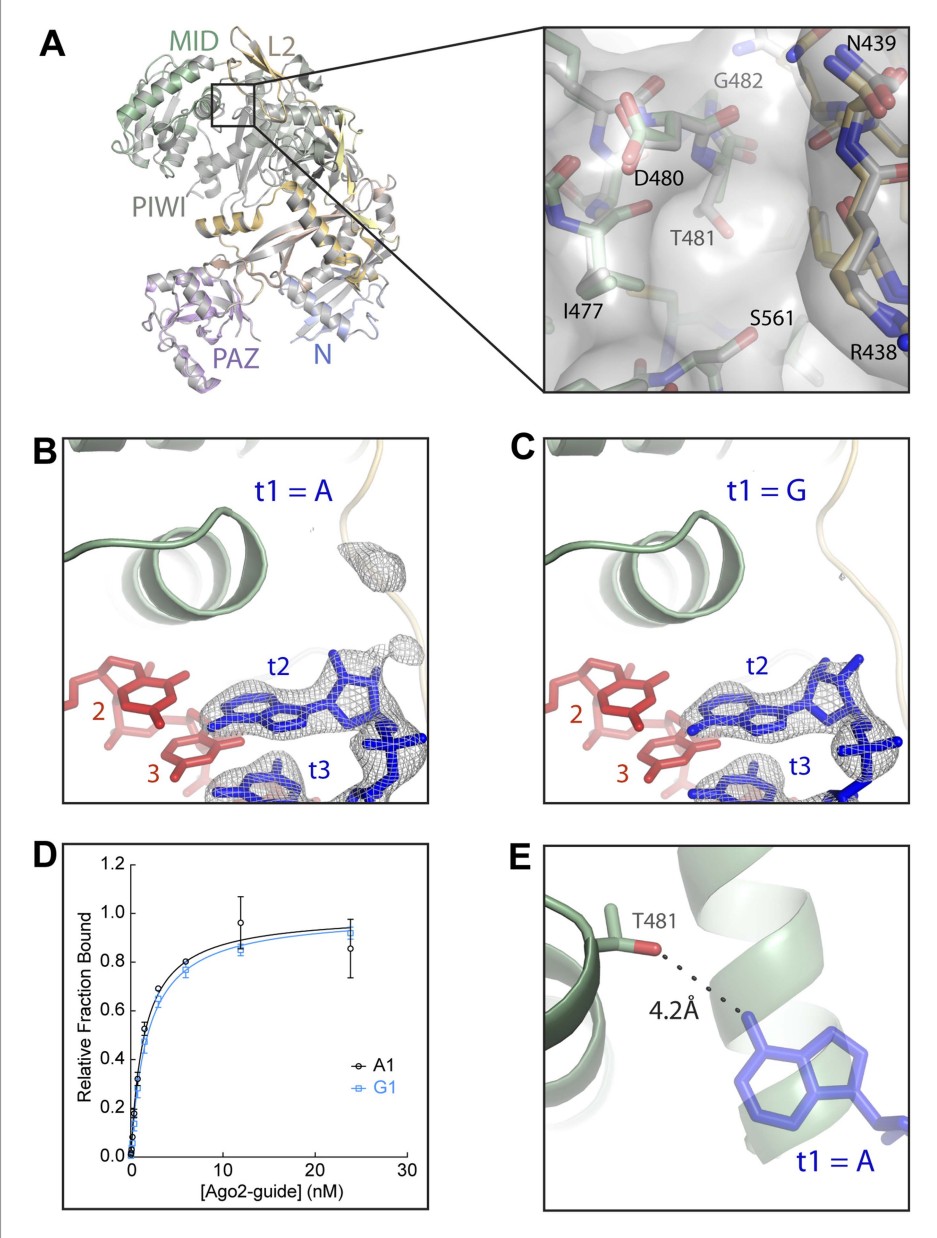

**Figure 5**. Disruption of the t1 pocket water network abolishes t1A recognition. (**A**) Overlay of wild-type and A481T Ago2 structures and close-up view of the region surrounding the A481T mutation (inset). (**B**, **C**) Crystal structures of A481T-Ago2 bound to t1A (**B**) or t1G (**C**) target RNAs. Target RNA omit maps contoured at 3σ (grey mesh). (**D**) Plot of bound t1A and t1G target RNAs vs A481T-Ago2-guide concentration. (**E**) Overlay of A481T-Ago2 and t1A from wild-type (WT) Ago2 (semi-transparent) structures. 4.2 Å distance between the T481 Oγ hydroxyl and the t1A N6 amine indicated as dashed line.

results suggest that t1A nucleotides are not used for initial target site recognition, but instead function primarily by enhancing the stability of the Ago2:target interaction after seed-pairing.

We note that, although the $\Delta\tau$ distribution of t1U is well described by a single-exponential decay ($\Delta\tau = 1.64 \pm 0.11$ s, $R^2 = 0.994$), the $\Delta\tau$ distributions of t1A and t1DAP deviate noticeably from ideal ($R^2 = 0.972$ and $0.961$ for t1A and t1DAP, respectively; *Figure 6E*, gray lines). This observation indicates that t1A and t1DAP binding events consist of at least two discrete populations. Assuming only two populations exist, we fit each data set to a double-exponential decay and calculated that about half of the binding events ($49.8\% \pm 6.2\%$ and $44.6\% \pm 4.4\%$ for t1A and t1DAP, respectively)

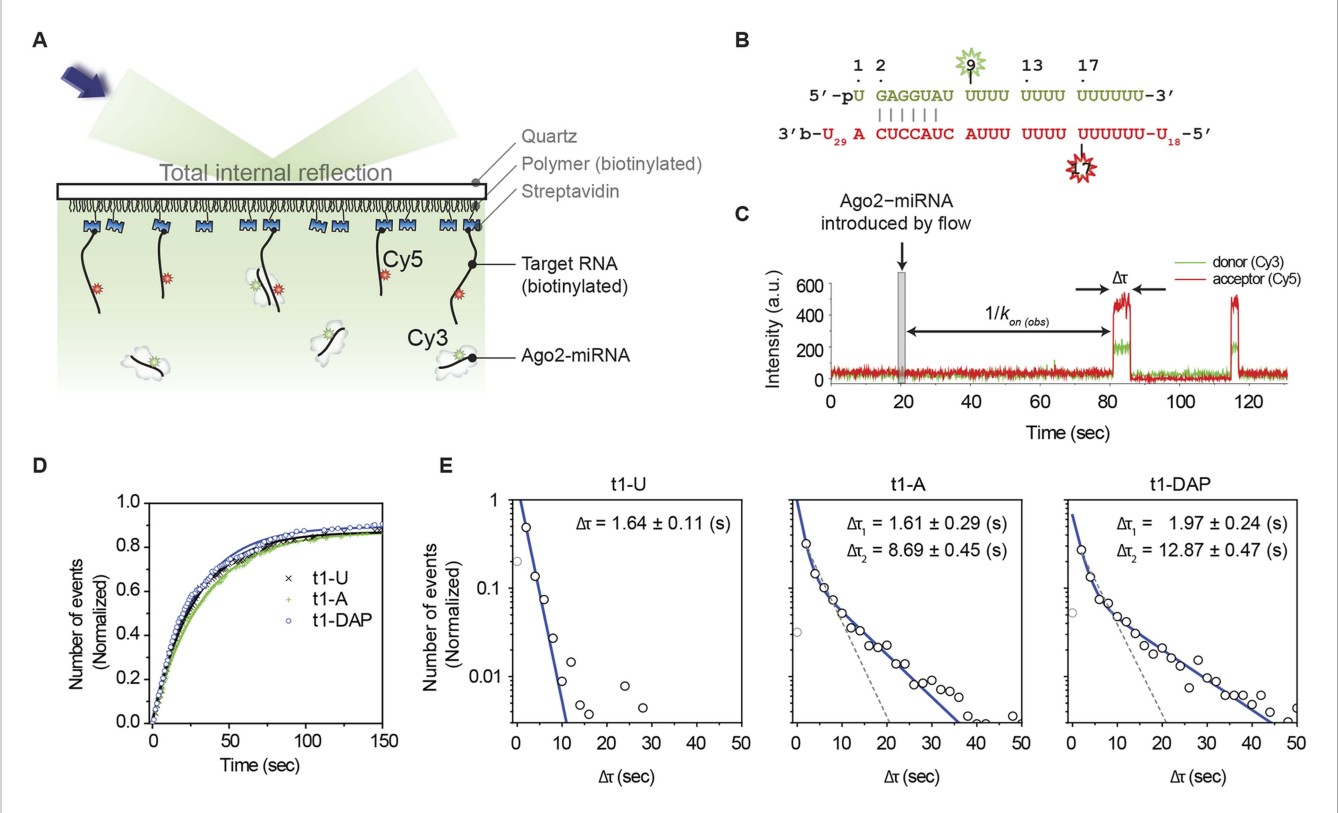

**Figure 6**. t1A nucleotides increase the dwell time on target sites. (**A**) Cartoon schematic of a single-molecule FRET assay. (**B**) Sequences of miRNA and target RNA with base pairs shown. The donor fluorophore (Cy3) is positioned on the ninth nt of miRNA (counting from the 5′ end of miRNA) and the acceptor (Cy5) on target RNA opposite nt 17 of miRNA. (**C**) Representative time trajectory. $\Delta\tau$, dwell time of interaction; $k_{on(obs)}$, apparent binding rate. The thin grey box indicates the time of a flow that delivers Ago2 and microRNA into the observation chamber. (**D**) Accumulated number of first Ago2-miRNA/target binding events vs time for RNA targets bearing t1U, t1A, or t1DAP. The number of events was normalized by the total number of target RNA strands over an imaging area. (**E**) Binding event dwell times fit to a double exponential decay (blue). The t1A binding events fit two populations ($49.8 \pm 6.2\%$ that exhibits $\Delta\tau_1$ and $50.2 \pm 6.2\%$ that exhibits $\Delta\tau_2$; $R^2 = 0.998$). The t1DAP binding events fit two populations ($44.6 \pm 4.4\%$ that exhibits $\Delta\tau_1$ and $55.4 \pm 4.4\%$ that exhibits $\Delta\tau_2$; $R^2 = 0.996$). Dotted grey lines represent fits to a single exponential decay. The first data columns were excluded to avoid artifacts arising from the time resolution limit. The number of events per bin was normalized by the total amount of binding events per each histogram.

had dwell times closely matching that of the t1U target ($\Delta\tau_1$ (t1A) = $1.61 \pm 0.29$ s; $\Delta\tau_1$ (t1DAP) = $1.97 \pm 0.24$). In contrast, the other half of the binding events had an average dwell time more than fivefold longer ($\Delta\tau_2$ (t1A) = $8.69 \pm 0.45$ s; $\Delta\tau_2$ (t1DAP) = $12.87 \pm 0.47$ s). The simplest explanation for these observations is that, when using limited seed-pairing as in our experimental setup, Ago2 dissociates from the target RNA without stably engaging t1A about half of the time. However, in cases in which Ago2 does engage t1A, the affinity of the interaction is increased substantially. This effect is exaggerated with t1DAP, which stabilizes the association further still. Notably, the t1A nucleobase inserts into the t1-binding pocket at an angle nearly orthogonal to the trajectory of the paired target RNA as it extends into the central cleft of Ago2 (*Figure 1*). Based on this observation, we suggest that t1A binding increases the dwell time on target RNAs by anchoring Ago2 to seed-paired target sites and inhibiting dissociation.

## Discussion

In this study we provide evidence that human Ago2 contains a t1-nucleotide binding pocket that specifically recognizes adenine nucleobases through water-mediated contacts. Our model is reminiscent of interactions in the *trp* operator/repressor complex, which also uses an ordered network of hydrogen-bonded water molecules to achieve nucleobase recognition (*Otwinowski et al., 1988*).

We also show that the adenosine N6 amine is a key determinant of specificity and that the addition of a methyl group to the N6 amine blocks t1A recognition. Taken with the finding that m6A modifications are enriched in mammalian mRNA 3′ UTRs (*Meyer et al., 2012*), this result raises the intriguing possibility that, in some cases, miRNA targets may be partially derepressed via adenosine methylation at 7mer-A1 and 8mer sites (*Bartel, 2009*).

Insights into t1 recognition by Ago2 can also be extended to other eukaryotic members of the Argonaute protein family. The amino acids making up the t1-binding site in Ago2 are conserved in all four human Argonaute proteins, and the structure of the t1-binding pocket in Ago2 is nearly identical to the same region in the structure of human Ago1 (*Faehnle et al., 2013; Nakanishi et al., 2013*). In fact, two ordered water molecules corresponding to waters A and B were observed in the structure of human Ago1 crystallized in the absence of target RNA (*Nakanishi et al., 2013*). We therefore suggest that all four human Argonautes likely recognize t1A nucleotides using the same mechanism. Additionally, recent studies indicate that some members of the Piwi clade of Argonaute proteins also display t1-nucleotide preferences. Specifically, the *Drosophila* protein Aubergine, the silkmoth protein Siwi, and the mouse protein Mili preferentially cleave targets bearing a t1A; while mouse protein Miwi2 prefers targets with a t1 purine (*Wang et al., 2014*). We note that most Piwi proteins displaying t1 preferences have conserved residues lining the t1-binding site (with the notable exception of Mili, which appears to lack several residues homologous to the t1A-binding residues in Ago2). In contrast, *Drosophila* and silkmoth Ago3, which do not display t1 preferences (*Wang et al., 2014*), lack homology in key residues making up the t1-binding pocket (*Figure 7*). In *C. elegans*, both t1A and t1U nucleotides are conserved features of many miRNA target sites, indicating that the Argonaute proteins Alg-1 and Alg-2 also have t1 nucleotide preferences (*Jan et al., 2011*). Curiously, there are only two minor differences between residues making up the human Ago2 t1-binding pocket and the equivalent residues in Alg-1 and Alg-2 (Asn-429 and Arg-479 in Ago2). Conceivably, these substitutions could reorganize the t1-pocket water network to recognize both adenine and uracil nucleobases. Alternatively, we suggest that Alg-1 and Alg-2 may specifically bind t1A, and that conservation of miRNA sites with t1U is connected to recognition by one or more of the other *C. elegans* Argonaute proteins.

Finally, we suggest that understanding the t1-binding site may provide new inroads for development of novel anti-miR oligonucleotides (*van Rooij and Kauppinen, 2014*). We found that the addition of a t1DAP nucleotide increased target affinity fourfold compared to non-t1A targets (*Table 2*). More importantly, the t1-binding pocket has a chemically diverse surface and is spacious

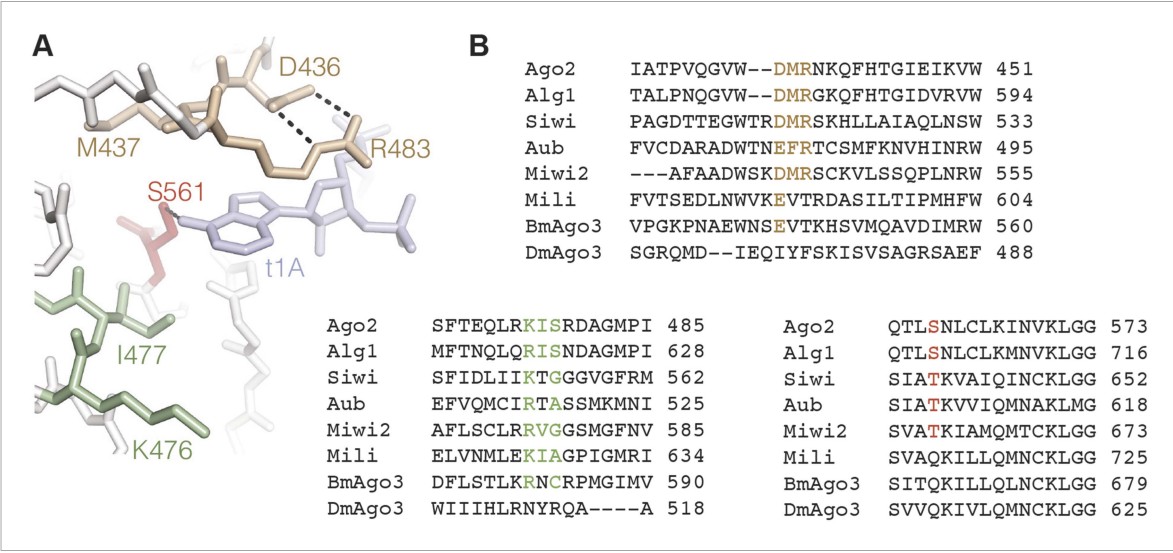

**Figure 7**. Conservation in the t1A-binding pocket. (**A**) Structure of the Ago2 t1A-binding pocket with major structural elements indicated. Ago2 shown as sticks with side chains of non-highlighted residues hidden for clarity. Target RNA shown in blue. (**B**) Multiple sequence alignment of Ago2 with other members of the extended Argonaute protein family. Conserved structural elements colored as in panel **A**.

enough to accommodate nucleotide analogues much more elaborate than DAP. Moreover, displacement of ordered water molecules A and B may lead to entropic gains that translate into higher binding affinity, in a fashion similar to that used in the development of cyclic urea HIV-1 protease inhibitors (*Lam et al., 1994*). We suggest that t1-nucleotide analogs that make favorable contacts inside the t1-binding pocket could lead to anti-miRs that specifically target the Argonaute-miRNA complex.

## Materials and methods

### Oligonucleotides

All RNA oligonucleotides were synthesized by Integrated DNA Technologies (IDT, Coralville, IA), with the exception of t1I and t1DAP, which were synthesized by ValueGene, and t1m6A, which was synthesized by GE Healthcare Dharmacon (Lafayette, CO). Prior to binding studies, all target RNAs were 5′ end labeled using $\gamma$-$^{32}$P-ATP and T4 polynucleotide kinase (New England Biolabs, Ipswich, MA) and subsequently purified by denaturing 16% polyacrylamide gel electrophoresis (PAGE) and ethanol precipitation.

### RNA nucleotide sequences

RNAs used in crystal structures and equilibrium binding experiments:
  *Guide RNA*: 5′ p-UUCACAUUGCCCAAGUCUCUU 3′;
  Target RNAs (t1 nucleotides in bold):

*t1A:* 5′ CAAUGUGA**A**AA 3′
*t1U:* 5′ CAAUGUGA**U**AA 3′
*t1C:* 5′ CAAUGUGA**C**AA 3′
*t1G:* 5′ CAAUGUGA**G**AA 3′
*t1I:* 5′ CAAUGUGA**I**AA 3
*t1DAP:* 5′ CAAUGUGA(**DAP**)AA 3′
*t1m6A:* 5′ CAAUGUGA(**m6A**)AA 3′

  RNAs used in single molecule experiments
  Guide RNA (U with C6 amino modifier for Cy3 attachment underlined): 5′ p-UGAGGUAU<u>U</u>UUU UUUUUUUUUU 3′

  Target RNAs (U with C6 amino modifier for Cy5 attachment underlined): *t1A:* 5′ (U)$_{20}$UUU<u>U</u>UU UUUUUACUACCUC**A**(U)$_{29}$-biotin 3′ *t1U:* 5′ (U)$_{20}$UUU<u>U</u>UUUUUUUACUACCUC**U**(U)$_{29}$-biotin 3′

  *t1DAP:* 5′ (U)$_{20}$UUU<u>U</u>UUUUUUUACUACCUC(**DAP**)(U)$_{29}$-biotin 3′

### Protein expression and purification

Full length wild type and A481T Ago2 proteins were expressed in Sf9 cells using a baculovirus system (*Schirle and MacRae, 2012*), and purified as described previously (*Schirle et al., 2014*). Briefly, His$_6$-tagged Ago2 was purified by Ni-chelate chromatography and loaded with single-stranded guide RNAs. Loaded Ago2 proteins where then isolated using the Arpón method for purifying Argonaute complexes loaded with a specified guide RNA (*Flores-Jasso et al., 2013*).

### Crystallization and data collection

Ago2-guide-target complexes were formed by mixing Ago2-guide complexes with target RNAs at a 1:1.2 molar ratio and incubating at room temperature for 10 min. Crystals were grown by hanging drop vapor diffusion at 20°C. Drops contained a 1:1 ratio of Ago2-guide-target to reservoir solution (16% PEG 3350, 0.1 M Tris, pH 8.0, 0.1 M phenol, 12% isopropanol, and 10 mM MgCl$_2$). Crystals typically appeared overnight, were harvested with nylon loops, and flash frozen in liquid N$_2$. Data were collected under cryogenic conditions remotely at beam line 12-2 at the Stanford Synchrotron Radiation Lightsource (SSRL), and beamline 24-ID-E at the Advanced Photon Source (APS) (*McPhillips et al., 2002*; *Soltis et al., 2008*). Data were processed using XDS and Scala (*Gonzalez and Tsai, 2010*; *Kabsch, 2010*; *Winn et al., 2011*).

### Structure refinement

All structures were refined using the Ago2-guide-target structure (PDB ID 4W5O), with the t1A nucleotide omitted, as a starting model. Models were built using Coot (*Emsley et al., 2010*) and were

subjected to XYZ coordinate, TLS, and B-factor refinement using PHENIX (*Adams et al., 2010*). Model building and refinement continued iteratively until all interpretable electron density was modeled. Water molecules were identified automatically in Coot ($2F_{obs} - F_{calc}$ map, above 1.8σ, and greater than 2.4 Å and less than 3.2 Å from hydrogen bond donors or acceptors) and by manual inspection of electron density maps. All structures were refined using an $R_{free}$ set identical to that used in refinement of the original 4W5O structure. Structure figures were generated with PyMOL (Schrödinger, LLC, Portland, OR).

## Equilibrium binding assays

Guide-loaded Ago2 samples (0–70 nM) were incubated with 0.1 nM $^{32}$P-labeled target RNAs in binding reaction buffer (30 mM Tris pH 8.0, 0.1 M potassium acetate, 2 mM magnesium acetate, 0.5 mM TCEP, 0.005% (vol/vol) NP-40, 0.01 mg/ml baker's yeast tRNA (Sigma, St. Louis, MO)), in a total reaction volume of 100 µl, for 45 min at room temperature. Protein–RNA complexes and free RNA were separated using a dot-blot apparatus (GE Healthcare Life Sciences, Pittsburgh, PA), using Protran nitrocellulose membrane (0.45 µm pore size, Whatman, GE Healthcare Life Sciences) to bind protein complexes, and Hybond Nylon membrane (Amersham, GE Healthcare Life Sciences) to capture free RNA. Samples were applied with vacuum and then washed with 100 µl of ice-cold wash buffer (30 mM Tris pH 8.0, 0.1 M potassium acetate, 2 mM magnesium acetate, 0.5 mM TCEP). Membranes were air-dried and $^{32}$P signal visualized by phosphorimaging. Quantification was performed using ImageQuant software (GE Healthcare Life Sciences), and dissociation constants calculated using Prism version 5.0e (GraphPad Software, Inc., La Jolla, CA).

## Single molecule FRET

Ago2 single molecule binding measurements were performed as described elsewhere (*Chandradoss et al., 2015*). Briefly, target RNAs bearing a Cy5 dye (GE Healthcare) and a 3′ biotin were immobilized on a polymer(PEG)-coated quartz surface in the microfluidic chamber (*Chandradoss et al., 2014*) of a prism-type total internal reflection fluorescence microscope. Ago2 was loaded with a guide miRNA containing a Cy3 dye (GE Healthcare). The resulting complex was introduced into the microfluidic chamber and Cy3 molecules were excited with a 532 nm diode laser (Compass 215M/50 mW, Coherent). Fluorescence signals of Cy3 and Cy5 were collected through a 60× water immersion objective (UplanSApo, Olympus, Center Valley, PA) with an inverted microscope (IX73, Olympus). Laser scattering was blocked by a 532 nm long pass filter (LPD01-532RU-25, Semrock, Rochester, NY). The Cy3 and Cy5 signals were separated with a dichroic mirror (635 dcxr, Chroma, Bellows Falls, VT) and imaged using a EM-CCD camera (iXon Ultra, DU-897U-CS0-#BV, Andor Technology, United Kingdom) and described previously (*Selvin and Ha, 2007*).

## Data acquisition and analysis

CCD images of time resolution 0.1 s were recorded, and time traces were extracted from the CCD image series using IDL (ITT Visual Information Solution, Boulder, CO). Colocalization between Cy3 and Cy5 signals was carried out with a custom-made mapping algorithm written in IDL. The extracted time traces were processed using Matlab (MathWorks, Natick, MA) and Origin (Origin Lab, Northampton, MA). The binding rate ($k_{on}$) was determined by first measuring the time between when Ago2-miRNA was introduced to a microfluidic chamber and when the first Ago2-miRNA was docked to a target RNA; and then fitting the time distribution with a single-exponential growth curve, $A(1 - e^{-k_{on}t})$. The dissociation rate was estimated by measuring the dwell time of a binding event. A dwell time distribution was fitted by either a single-exponential decay curve ($Ae^{-t/\Delta\tau}$) or a double-exponential decay curve ($A_1 e^{-t/\Delta\tau_1} + A_2 e^{-t/\Delta\tau_2}$). In case of a double-exponential decay, the percentages of $\Delta\tau_1$ and $\Delta\tau_2$ populations are determined by $A_1\Delta\tau_1/(A_1\Delta\tau_1 + A_2\Delta\tau_2)$ and $A_2\Delta\tau_2/(A_1\Delta\tau_1 + A_2\Delta\tau_2)$, and the average dwell time is determined by $(A_1\Delta\tau_1{}^2 + A_2\tau_2{}^2)/(A_1\Delta\tau_1 + A_2\Delta\tau_2)$.

## Single molecule sample preparation

A microfluidic chamber was incubated with 20 µl Streptavidin (0.1 mg/ml, Sigma) for 30 s. Unbound Streptavidin was washed with 100 µl of buffer T50 (10 mM Tris–HCl [pH 8.0], 50 mM NaCl buffer). The 50 µl of 50 pM acceptor-labelled mRNA construct were introduced into the chamber and incubated for 1 min. Unbound labeled constructs were washed with 100 µl of buffer T50. The effector complex was formed by incubating 50 nM purified recombinant hAgo2 with 0.5 nM of donor-labeled hsa-let-7a

miRNA in a buffer containing 50 mM Tris–HCl [pH 8.0] (Ambion, Grand Island, NY), 50 mM NaCl (Ambion), and 60 mM KCl (Ambion) at 31°C for 20 min. An imaging buffer for single-molecule FRET was added before the mixture was injected to a microfluidic chamber. The final concentration of the imaging buffer consists of the 0.8% dextrose (Sigma), 0.5 mg/ml glucose oxidase (Sigma), 85 µg/ml Catalase (Merck), and 1 mM Trolox ((±)-6-hydroxy-2,5,7,8-tetramethylchromane-2-carboxylic acid, 238813, Sigma). The experiments were performed at the room temperature (23 ± 2°C).

## Acknowledgements

We are grateful to JR Williamson, DP Bartel, JM Claycomb, E Westhof and JS Oakdale for generous advice and insights. Diffraction data were collected at beam lines: 11-1 and 12-2 at the Stanford Synchrotron Radiation Lightsource, supported by the U.S. Department of Energy, Office of Science, Office of Basic Energy Sciences under Contract No. DE-AC02-76SF00515; 5.0.2 at the Advanced Light Source, supported by the Director, Office of Science, Office of Basic Energy Sciences, of the U.S. Department of Energy under Contract No. DE-AC02-05CH11231; and, 24-ID-E at the Advanced Photon Source, supported by NIGMS grant P41 GM103403 and DOE contract number DE-AC02-06CH11357. NTS and JSG are pre-doctoral fellows of the American Heart Association. NTS is an Achievement Rewards for College Scientists (ARCS) scholar. CJ was funded by the Open Program of the Division for Earth and Life Sciences (822.02.008) of the Netherlands Organization for Scientific Research and European Research Council under the European Union's Seventh Framework Programme [FP7/2007–2013]/ERC grant agreement no. [309509]. The work was supported by NIH grant R01 GM104475 to IJM. Atomic coordinates for the following structures (with PDB IDs) have been deposited in the PDB: wild-type Ago2 bound to target RNAs bearing t1C (4Z4C), t1G (4Z4D), t1U (4Z4E), t1DAP (4Z4F), t1I (4Z4G); and, A481T Ago2 bound to target RNAs with t1A (4Z4H) and t1G (4Z4I).

## Additional information

### Funding

| Funder | Grant reference | Author |
| --- | --- | --- |
| National Institute of General Medical Sciences (NIGMS) | R01-GM104475, P41-GM103403 | Ian J MacRae |
| European Research Council (ERC) | 309509 | Chirlmin Joo |
| U.S. Department of Energy (DOE) | DE-AC02-76SF00515, DE-AC02-05CH11231, DE-AC02-06CH11357 | Ian J MacRae |
| American Heart Association (AHA) | Graduate Student Fellowship | Nicole T Schirle, Jessica Sheu-Gruttadauria |

The funders had no role in study design, data collection and interpretation, or the decision to submit the work for publication.

### Author contributions

NTS, Conception and design, Acquisition of data, Analysis and interpretation of data, Drafting or revising the article; JS-G, SDC, Conception and design, Acquisition of data, Analysis and interpretation of data; CJ, IJMR, Conception and design, Analysis and interpretation of data, Drafting or revising the article

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
