## [Decision Letter]

Thank you for submitting your work entitled “Water-mediated recognition of t1-adenosine anchors Argonaute2 to microRNA targets” for peer review at *eLife*. Your submission has been favorably evaluated by John Kuriyan (Senior Editor), and three reviewers, one of whom is a member of our Board of Reviewing Editors.

The following individuals responsible for the peer review of your submission have agreed to reveal their identity: Phillip Zamore (Reviewing Editor), Yukihide Tomari, and David Bartel (peer reviewers).

The reviewers have discussed the reviews with one another and the Reviewing editor has drafted this decision to help you prepare a revised submission.

Despite the presence of a complementary guide g1 nucleotide, the t1 target nucleotide is typically recognized by Argonaute proteins rather than base pairing. In mammals, miRNA target sites often have an adenosine (A) across from the first nucleotide of the guide miRNAs. Such a “t1A” preference is observed also for a subset of PIWI proteins in flies and mice. Recent crystal structures of human Ago2 revealed the existence of a surface pocket at the interface of the L2 linker and the MID domain, which can accommodate the t1 nucleotide. However, the pocket is apparently too large to discriminate among the four nucleotides, so it has been unclear why t1A is specifically preferred. In this manuscript, Schirle et al. report the structures of human Ago2 bound to target RNAs with various nucleotides at the t1 position. MacRae and colleagues find report that a network of water molecules in the t1A pocket provides the specificity for recognizing adenine, especially via its N6 amine. Using single-molecule imaging, the authors suggest that the t1A nucleotide is not used for the initial association with target, but instead slows dissociation of RISC from the target. Finally, the authors identify a base analog, di-aminopurine, that binds more tightly than adenine, leading them to suggest that novel bases could be designed to provide additional specificity to anti-miRNA drugs. The manuscript would be improved by addressing the points below, but otherwise is entirely appropriate for publication in *eLife*.

1) “[S]liding off of target sites” implies a specific physical model in which Ago2 departs from its seed-matched binding site but not the RNA itself. Does the evidence really support this?

2) In the determination of the *k*_*off*_ for t1A, a test statistic or some other analysis (e.g., residuals) should be provided showing the statistical confidence that the data are better explained by a double exponential than a single exponential. If a double-exponential is, in fact, justified, what is the meaning of the t1A single-molecule data fitting a double exponential? That is, what is the physical basis for the 16% of RISC that is slow-to-dissociate? Might it be a non-biological interaction caused by the slide or its coating?

3) Given that the water molecules are critical in the wild-type t1A structure, the omit map for the four water molecules should be shown (for Figure 1 or 4A). The authors should also note whether the four water molecules are observed in the non-t1A structures. In other words, is the hydrogen-bonding network of water molecules formed independently of or in coordination with t1A binding?

4) At the end of the subsection “A purine N6 amine is required for t1 recognition”, the authors state that the N2 amine has a positive effect on t1 binding but do not describe what kind of interactions are made between the N2 amine and Ago2. Do the authors observe the similar interactions for the N2 amine of t1G? Also, it would be timely for the authors to comment on how 6-methyl adenine fits into the t1A pocket compared to adenine.

5) At the end of the subsection “Disruption of the water network extinguishes t1-A recognition“ and in Figure 5, it is difficult to see the positional relationship between T481 and t1A, which probably could be better shown in a separate figure.

6) It would be good to test by single-molecule analysis that the A481T mutant and/or t1-Inosine do not increase the dwell time.

7) The *k*_*on*_ and *k*_*off*_ values from the single-molecule data in Figure 6 result in KD values 50-fold higher than those determined from the ensemble assays (Table 2). To prevent confusion, the authors should acknowledge this difference and state whether they think it is due to the different sequences used in the two types of experiments or to the Cy3 on position 9 of the miRNA used for the single-molecule experiments.

8) It is not apparent how the behavior depicted in Figure 6 is used to calculate the *k*_*on*_ for RISC-target association. The concentration of RISC in the microfluidic chamber should be provided in the manuscript, as well as a more detailed description of the method by which *k*_*on*_ is calculated.

9) The authors should comment on the consistency between the ensemble (Table 2) and the single-molecule data with respect to the double-exponential interpretation of t1A dissociation kinetics. Specifically, the authors should calculate the effects on the ensemble *k*_*off*_ for t1A predicted from the single-molecule experiments and verify that the ratio of *k*_*off*_ values with t1A and t1U targets is agrees with the ratio of KD values for t1A and t1U targets. (These numbers appear to work out by one Reviewer's calculations, but it would be useful to show that the two methods are in agreement.)

10) Can the authors explain why <40% of target RNAs are bound by Ago2-RISC in Figure 6? What is the specific activity of Ago2-RISC used in this study?

11) The photobleaching time should be shown for Cy3 alone, Cy5 alone and FRET.

12) A multiple alignment of Ago family proteins would help the discussion of t1A-binding pocket conservation.

---

## [Author Response]

*Despite the presence of a complementary guide g1 nucleotide, the t1 target nucleotide is typically recognized by Argonaute proteins rather than base pairing. In mammals, miRNA target sites often have an adenosine (A) across from the first nucleotide of the guide miRNAs. Such a “t1A” preference is observed also for a subset of PIWI proteins in flies and mice. Recent crystal structures of human Ago2 revealed the existence of a surface pocket at the interface of the L2 linker and the MID domain, which can accommodate the t1 nucleotide. However, the pocket is apparently too large to discriminate among the four nucleotides, so it has been unclear why t1A is specifically preferred. In this manuscript, Schirle et al. report the structures of human Ago2 bound to target RNAs with various nucleotides at the t1 position. MacRae and colleagues find report that a network of water molecules in the t1A pocket provides the specificity for recognizing adenine, especially via its N6 amine. Using single-molecule imaging, the authors suggest that the t1A nucleotide is not used for the initial association with target, but instead slows dissociation of RISC from the target. Finally, the authors identify a base analog, di-aminopurine, that binds more tightly than adenine, leading them to suggest that novel bases could be designed to provide additional specificity to anti-miRNA drugs. The manuscript would be improved by addressing the points below, but otherwise is entirely appropriate for publication in eLife*.

We are grateful to the reviewers for their interest and careful evaluation of our work. We have revised the manuscript to address all of the criticisms and suggestions. The revised paper includes new data showing that adenosine N6 methylation abrogates t1A recognition by Ago2, and that the N2 purine amine extends average dwell time of the Ago2-guide complex on target RNA sites. These findings support and extend conclusions stated in the original submission.

1) “[S]liding off of target sites” implies a specific physical model in which Ago2 departs from its seed-matched binding site but not the RNA itself. Does the evidence really support this?

This point is well taken. We changed “sliding off of target sites” to “dissociating from target sites”.

*2) In the determination of the* k_off_
*for t1A, a test statistic or some other analysis (e.g., residuals) should be provided showing the statistical confidence that the data are better explained by a double exponential than a single exponential*.

The calculated coefficients of determination for the *k*_*off*_ data (now indicated in the main text) are as follows, (Note: we have included new data for t1DAP binding): t1U: R^2^ = 0.994 (single exponential) t1A: R^2^ = 0.972 (single); 0.998 (double) t1DAP: R^2^ = 0.961 (single); 0.996 (double)

Although all data were collected in the same way, the distribution of t1A and t1DAP dwell times do not fit a single exponential as well as the t1U data do. To better illustrate this point, we displayed dwell time distributions as semi-log plots, in which t1A and t1DAP both clearly deviate from the linear relationship expected of a single exponential function (see updated Figure 6).

If a double-exponential is, in fact, justified, what is the meaning of the t1A single-molecule data fitting a double exponential? That is, what is the physical basis for the 16% of RISC that is slow-to-dissociate? Might it be a non-biological interaction caused by the slide or its coating?

A surface artifact is unlikely because the slow-to-dissociate population was not observed with t1U, which was tested on the same slide (but in a different chamber) as t1A. The simplest explanation is that the two populations represent binding events in which Ago2 either does or does not stably engage t1A before dissociating. This model is supported by: 1) the observation that the dwell time of the short-lived t1A binding events (Δτ_1_ = 1.61 sec) closely matches the dwell time of t1U (Δτ = 1.64 sec); and, 2) the long-lived dwell time (Δτ_2_ = 8.69 sec) can be increased further still by replacing t1A with t1DAP (Δτ_2_ = 12.87 sec); and, 3) t1A and t1DAP do not influence on-rates, demonstrating that their effects on affinity manifest after seed-pairing.

Separately, we note that in the first submission we used incorrect formulas for estimating the percentages of Δ*τ*_1_ and Δ*τ*_2_ populations within a double exponential decay. Specifically, in the initial submission, we used wrong equations for the percentages, *A*_1_/(*A*_1_ + *A*_2_) and *A*_2_/(*A*_1_ + *A*_2_), where *A*_1_ and *A*_2_ are the coefficients of exponentials. This estimate resulted in 16% for the slow-to-dissociate population. With correct equations, *A*_1_Δ*τ*_1_/*A*_1_Δ*τ*_1_ + *A*_2_Δ*τA* and *A*_2_*τ*_2_/*A*_1_Δ*τ*_1_ + *A*_2_Δ*τ*_2_, we calculate that the slow-to-dissociate population constitutes ∼50% of the binding events. This change does not alter our interpretation of the slow-to-dissociate population. Rather, it indicates that the t1A influences the dissociation pathway to a larger extent than we initially estimated. The correct formulas are described in a new section in the Materials and methods.

*3) Given that the water molecules are critical in the wild-type t1A structure, the omit map for the four water molecules should be shown (for*
Figure 1
*or 4A). The authors should also note whether the four water molecules are observed in the non-t1A structures. In other words, is the hydrogen-bonding network of water molecules formed independently of or in coordination with t1A binding?*

As suggested, we added a water-omit map calculated from the t1-A dataset to Figure 4. We also note that ordered water molecules corresponding to waters A, B and D are clearly observed in the t1-U electron density map. Taken with the observation of water molecules equivalent to A and B in the crystal structure of Ago1 in the absence of target RNA (Nakanishi, et al., Cell Reports 2013; PDB ID 4KXT), it appears that Ago organizes t1-binding pocket water molecules, at least to some extent, prior to t1-A binding.

4) At the end of the subsection “A purine N6 amine is required for t1 recognition”, the authors state that the N2 amine has a positive effect on t1 binding but do not describe what kind of interactions are made between the N2 amine and Ago2. Do the authors observe the similar interactions for the N2 amine of t1G?

The N2 amine does not directly contact Ago2 and thus it is difficult to explain the observed increases in dwell time and affinity. Our currently favored explanation is that the N2 amine may make positive interactions with water C, possibly moving it into hydrogen-bonding distance of water B. This idea is based on a positive difference density peak, which may correspond to the repositioned water C, observed in the t1-G data (Figure 8). We speculate that a similar change in solvent position may occur with t1-DAP binding. Unfortunately, the t1-DAP dataset is relatively low resolution (2.8 Å) and we were not able to model any water molecules in the t1-pocket with confidence. Without direct observation of repositioned water molecules in the t1-DAP structure, we are hesitant to put this model forward and thus simply added the sentence: “Waters “C” and “D” likely provide an additional layer of selectivity through interactions with the unprotonated N1 amine on t1-A, and may interact with the purine N2 amine, which does not directly contact Ago2” to state that N2 does not contact Ago2”.

Author response image 1.Electron density maps surrounding the t1-G nucleotide.2Fo-Fc map is shown contoured at 1σ(blue mesh). Fo-Fc map (difference map) is shown contoured at 3σ (green mesh) and -3σ (red mesh). A positive difference density peak, possibly representing a repositioned water C, is observed 2.5 Å from water B and 2.7 Å from the t1-G N2 amine. Position of water C in t1-A structure is included for comparison.**DOI:**
http://dx.doi.org/10.7554/eLife.07646.015

*Also, it would be timely for the authors to comment on how 6-methyl adenine fits into the t1A pocket compared to adenine*.

We thank the reviewers for this thoughtful suggestion and decided the issue important enough to warrant determining experimentally how adenosine N6 methylation affects target binding. We found that a t1-m6A target RNA had an affinity similar to that of equivalent non-t1-A targets (see updated Figure 3 and Table 2), indicating that the modification effectively blocks t1-A recognition. This result further supports the idea that the N6 amine is a key determinant in t1-A recognition and raises the intriguing possibility that adenosine methylation could lead to partial derepression of miRNA targets containing 7mer-A1 or 8mer sites.

*5) At the end of the subsection “Disruption of the water network extinguishes t1-A recognition“ and in*
Figure 5*, it is difficult to see the positional relationship between T481 and t1A, which probably could be better shown in a separate figure.*

We apologize for this shortcoming and added panel E to Figure 5 to specifically illustrate the spatial relationship between T481 and t1A.

*6) It would be good to test by single-molecule analysis that the A481T mutant and/or t1-Inosine do not increase the dwell time*.

We thank the reviewers for this thoughtful suggestion. We conducted a slightly different experiment and measured association/dissociation rates of a target RNA with t1-DAP, with dual purposes of further aligning the bulk and single molecule studies, as well as providing insights into why t1-DAP binds better than t1-A. Consistent with the bulk studies, single molecule measurements showed an increase in affinity for t1-DAP over t1-A. The increase in affinity comes entirely from an increase in the average dwell time, with the on-rate unaffected by the additional N2 amine. These results support the notion that t1-nucleotides are not used in the initial target search.

*7) The* k_on_
*and* k_off_
*values from the single-molecule data in*
Figure 6
*result in KD values 50-fold higher than those determined from the ensemble assays (*Table 2*). To prevent confusion, the authors should acknowledge this difference and state whether they think it is due to the different sequences used in the two types of experiments or to the Cy3 on position 9 of the miRNA used for the single-molecule experiments*.

We thank the reviewers for identifying this potential point of confusion. The target RNAs used in the ensemble and the single molecule experiments are different from each other in several respects. The ensemble assays used very short target RNAs (11 nt) that pair to guide nt 2–9, which provided the best diffracting crystals. Our rational was that using the same RNAs for binding experiments and structure determination would be the best way to correlate structural findings with affinity measurements. The single molecule experiments, on the other hand, required longer target RNAs (69 nt) for reliable photophysics (to establish an optimum distance between donor and acceptor dyes) that paired to guide nt 2–7 (to work in an affinity range where dwell times could be accurately measured). We added a footnote to the subsection “t1A interactions increase the dwell time of Ago2 on miRNA target sites” to direct the reader to these differences.

We also note that, despite substantially different experimental setups and differences in target RNA sequence, length and guide-complementarity, all measured affinities fall within the low nM range. More importantly (for the focus of this study), the relative effects of t1-nucleotide identity are consistent between diverse experiments, and thus appear to be largely independent of these factors.

*8) It is not apparent how the behavior depicted in*
Figure 6
*is used to calculate the k*_*on*_
*for RISC-target association. The concentration of RISC in the microfluidic chamber should be provided in the manuscript, as well as a more detailed description of the method by which k*_*on*_
*is calculated*.

We are sorry for this oversight. We added new sections to the Materials and methods that are dedicated to explaining the experimental conditions (Single-molecule sample preparation) and the data analysis (Data acquisition and analysis).

*9) The authors should comment on the consistency between the ensemble (*Table 2*) and the single-molecule data with respect to the double-exponential interpretation of t1A dissociation kinetics. Specifically, the authors should calculate the effects on the ensemble* k_off_
*for t1A predicted from the single-molecule experiments and verify that the ratio of* k_off_
*values with t1A and t1U targets is agrees with the ratio of KD values for t1A and t1U targets. (These numbers appear to work out by one Reviewer's calculations, but it would be useful to show that the two methods are in agreement*.*)*

The average dwell times of t1U and t1A are 1.64 and 5.17 sec, respectively (for calculations see equations in Data acquisition and analysis). The ratio between <Δτ^t1U^> and <Δτ^t1A^> is therefore 3.2, which is similar to the ratio between *K*_D_^t1U^ and *K*_D_^t1A^, (2.5). Thus, the effects of t1A are relatively consistent between ensemble and single-molecule data.

*10) Can the authors explain why <40% of target RNAs are bound by Ago2-RISC in*
Figure 6*? What is the specific activity of Ago2-RISC used in this study?*

We thank the reviewers for this careful observation and have repeated all the single molecule measurements using a new preparation of protein and higher sample concentrations (originally 10 nM Ago2 mixed with 1 nM RNA guide, and now 50 nM Ago2 mixed with 0.5 nM guide RNA) to assemble RISC. Under the new conditions we could observe that the percentage of occupied target RNA reaches nearly 100%. We note that the binding rates we observe also increased, which might be due to the increase of a functional RISC concentration in the new sample preparation.

*11) The photobleaching time should be shown for Cy3 alone, Cy5 alone and FRET*.

We measured photobleaching of the donor Cy3 dye, which has a half time (394 seconds) > 20x longer than the average dwell times of the observed binding events (Figure 9). Cy5 and FRET photobleaching time is not necessary to measure since it is only Cy3 signals that are used for measuring *k*_*on*_ and *k*_*off*_.

Author response image 2.Photobleaching of the Cy3 donor.**DOI:**
http://dx.doi.org/10.7554/eLife.07646.016

*12) A multiple alignment of Ago family proteins would help the discussion of t1A-binding pocket conservation*.

We made a sequence alignment of Ago2 with Alg1 and Piwi proteins and related it to major structural elements of the t1-binding pocket in a new Figure 7.